# Neuromuscular Response during Different Side-Cutting Maneuvers and Its Influence on the Risk of Knee Injuries

**DOI:** 10.3390/sports11100190

**Published:** 2023-10-03

**Authors:** Adrián Feria-Madueño, Timothy E. Hewett, Alejandro Sánchez-Arteaga, Borja Sañudo

**Affiliations:** 1Department of Physical Education and Sport, University of Seville. Pirotecnia Street, 41013 Seville, Spain; aferia1@us.es; 2Biomechanics Laboratories and Sports Medicine Research, Department of Orthopedics, Marshall University, Huntington, WV 25755, USA; hewettt@marshall.edu; 3Hewett Global Consulting, Newport, KY 25701, USA; 4Department of General Surgery, Hospital Universitario Virgen del Rocío, 41013 Seville, Spain; alejandros.arteaga@gmail.com

**Keywords:** side-cutting, injury prevention, knee, neuromuscular control, exercise program

## Abstract

Background: The objective of this study was to investigate the impact of a customized exercise program on electromyographic (EMG) variables during side-cutting maneuvers. Methods: Fifty-seven physically active volunteers were recruited and randomized into two groups: a control group (CG) and an experimental group (EG). The CG maintained their regular physical activity, while the EG followed an individualized, integrated exercise regimen that included strength, neuromuscular, proprioceptive, eccentric training, and whole-body vibration (WBV) exercises, lasting for 12 weeks. EMG-Root Means Square (RMS) values for the quadriceps (Q) and hamstrings (H) were measured immediately following initial foot contact at 50, 100, 150, and 200 milliseconds. Changes in EMG activity were evaluated before and after a twelve-week intervention, and the effect size (ES, 90% confidence limit [90%CL]) was calculated. Results: The results showed that the EG participants exhibited enhanced co-contraction as measured by the hamstring/quadriceps ratio (H/Q ratio) during both open and cross-cutting side maneuvers after the twelve-week intervention. Conclusion: Furthermore, the customized, integrated exercise program that combined strength, neuromuscular, proprioceptive, eccentric training, and WBV exercises were could potentially serve as a beneficial neuromuscular and biomechanical strategy for addressing knee injury risk in non-professional, physically active populations at high risk for such injuries.

## 1. Introduction

Side-cutting is a crucial movement associated with the risk of lower limb injuries [1], particularly knee injuries [2]. The anterior cruciate ligament (ACL) is particularly vulnerable during cutting maneuvers as it restricts tibial translation [3], making it a significant risk factor for knee injuries. This movement is common in various professional sports such as football, basketball, rugby, hockey, and American football [4], and recent studies have reported an increased risk of knee injuries in recreational athletes during side-cutting [5].

There are numerous risk factors associated with knee injuries during cutting actions. Recently, Nedergaard et al. [6] identified biomechanical and neuromuscular factors as the primary risk factors for knee injuries during side-cutting maneuvers. They found that electromyography (EMG) variables were strongly associated with ACL injuries risk. EMG measurements of muscle activation have been suggested as potential outcome markers for knee injuries sustained during side-cutting maneuvers [7]. In any case, the role of muscle activation on the load supported by the ACL is crucial. When the knee is in extension, the shear force generated can lead to an increase in injury risk [8]. The musculature of the lower limb plays a crucial role in supporting the loads exerted on the knee joint, serving as a containment mechanism during tasks that involve side-cutting and landing maneuvers [8,9]. In recent years, scientific literature has pointed out that improved activation of the hamstring muscles may be beneficial for controlling tibial translation, which is associated with a reduction in the risk of injury. The co-activation ratio between the hamstrings (H) and quadriceps (Q) has been shown to play a critical role in preventing knee injuries. A systematic review reported that an H/Q ratio below 0.8 is associated with an increased risk of injury [10].

While numerous studies have explored EMG activity during cutting maneuvers [11,12], there is a need for an intervention that focuses exclusively on implementing a training program to modify EMG patterns during side-cutting maneuvers in non-professional athletes, without relying on isokinetic actions [7].

Various exercise programs have been proposed to improve neuromuscular control and joint stability and reduce the probability of knee injuries. For instance, Ribeiro-Alvares et al. [13] reported a reduced injury risk after exercises with eccentric components, such as the Russian/Nordic Hamstring Exercise, which targets both the H and Q muscles. It has also been reported that exercise programs that increase EMG activity of the hamstrings can reduce the risk of injury [14]. In an important systematic review and meta-analysis, Pappas et al. [15] demonstrated that physical exercise programs are linked to an improvement in neuromuscular deficits associated with knee injuries during cutting tasks. Blasimann et al. [16] have suggested that the assessment of neuromuscular control using EMG may be a crucial factor in determining the readiness of individuals with knee injuries to return to sports participation. More recently, the use of EMG as a tool to monitor muscle activation changes in individuals who have undergone ACL reconstruction has also been reported [17]. However, there is a lack of research that has investigated the potential preventative effects of sports training programs on EMG variables in young recreational athletes during different types of side-cutting maneuvers. Therefore, the purpose of the current study was to examine the impact of a tailored and integrative (combined) neuromuscular training exercise program on EMG activity during side-cutting activities in young recreational athletes. Furthermore, this explorative study aimed to investigate the effect of a training intervention program on modifying the co-activation ratio of the thigh musculature to approach a value of 1, as well as increasing poor H muscle activations during side-cutting.

## 2. Materials and Methods

### 2.1. Subjects

A sample of fifty-seven recreational athletes (13 female, 44 male) were recruited voluntarily for the study. To be included, participants were required to be non-elite athletes and engaged in a systematic sports practice three times a week. The sport practiced were handball, soccer, paddle tennis, athletics, and running. Participants who have sustained an injury in lower limbs that prevented them from performing the test protocol or in the prior 12 months were excluded. No subject was able to exercise 48 h before the test. Moreover, the researchers interviewed the participants to obtain information on the types of injuries previously sustained, as well as the experience in sports practice (i.e., duration and intensity). Verbal information about the aims and procedures were given to the participants, and they signed the approved consent form prior to the test. The study was approved by the Ethics Committee of University of Seville and was in accordance with the Declaration of Helsinki.

### 2.2. Experimental Design

The effect of a specific exercise program on EMG variables during side-cuttings maneuvers was determined with a highly structured experimental design. Participants completed three Maximal Voluntary Contraction (MVC) trials in the laboratory (Biodex Multi-joint System, Shirley, NY, USA). Following the testing measurements, participants underwent a familiarization session which involved performing 8–10 side-cuttings of each type. One day later, subjects completed three side-cutting tests and EMG-lower limb was evaluated. Volunteers were randomly allocated to experimental group (EG, n:29) or control group (CG, n: 28). The randomization sequence and blinding were generated using flipping a coin (heads-EG, trials-CG) [18]. A sample size was calculated by *t*-test for differences between independent groups (two groups), with an effect size of 0.8, an ɑ of 0.05 and a power of 0.80. The required sample was 26 for both groups. The calculation was performed based on data from a pilot study using G*power (v3.1, Heinrich-Heine-University, Dusselford, Germany).

### 2.3. Procedures

A standardized warm-up was developed, consisting of five minutes on a cycloergometer (Ergoline 900, Ergometrics, Bitz, Germany) at 60 W (60 rpm). Additionally, participants completed the warm-up session by performing 8–10 side-cuttings, similar to those in the familiarization session. The side-cuttings included angles of 30 degrees (SC30) and 45 degrees (SC45), as well as crossover movements at 45-degree angles (SC45cl). The last foot contact was recorded using a force plate (Kistler 9260 AA6, Winterthur, Switzerland). Subsequently, four tapes were placed on the ground to indicate the desired running directions (Figure 1). Participants were instructed to anticipate the start of each trial, while their running speed was controlled using a metronome set between 4 and 5.5 m/s [1].

When the baseline assessment was completed, the participants were randomly assigned to two groups. Over the course of 12 weeks, both groups continued with their regular sports activities, while the experimental group (EG) performed a customized exercise program three times a week. It consisted of strength, eccentric, proprioceptive, whole-body vibration (WBV) and neuromuscular exercise. The exercise program was designed by experienced diplomaed sports scientists, who adjusted the training loads according to the principle of the progression and individualization. In the first three weeks, the time of the training was between 40 and 45 min. Exercises with WBV, proprioception, eccentric strength, suspension training and balance were contemplated. In the fourth, fifth, and sixth week the same exercise group was executed but mixing its components (i.e., WBV and balance) in a total session with 50 min duration. For the three next weeks, crossing jumps and coordination task were added to the program. The total exercises and the number of the sets and repetitions was increased, bringing the total time to 60 min. Finally, the last three side-cutting tasks, stop tasks, and turns were added to the program. After 12 weeks, both EG and CG replicated the same tests in the laboratory under identical evaluation conditions (Appendix A).

### 2.4. Isometric Testing

The three MVC tests were evaluated with a prone position with 45-degree knee flexion to measure the hamstring muscles [19]. A 45-degree knee extension was determined during seated position for quadriceps muscles [20]. A 30 s rest was given between trials to reduce the fatigue [21]. Participants limited trunk movement by means of a harness across the subject’s abdomen. The MVC test was developed by aligning system and dominant leg, facilitating mechanical action on an anteroposterior axis motion. For 5 s, the researchers provided verbal feedback to maximally complete the isometric actions.

### 2.5. Electromyography

The EMG signal was evaluated on the dominant limb [22]. In this regard, the skin area was cleaned and disinfectant was applied with 96° isopropyl alcohol and cotton. The skin was shaved to allow the adhesion of the electrodes in the specific area of the skin. Subsequently, self-adhesive electrodes (diameter: 1.5 cm, distance between electrodes 3 cm; Blue Sensor, Medicotest A/S, Olstykke, Denmark) were placed after palpating the belly of the muscle while there was an isometric contraction. The reference electrode was placed over the patella and a wireless system was used to acquire the signals (bioPLUX, Lisbon, Portugal). EMG signal recording was performed following the protocol described by the European Project “Surface ElectroMyoGraphy for the Non-Invasive Assessment of Muscles” (SENIAM: http://www.seniam.org, accessed on 29th September 2023). A band-stop filter of 2 Hz centered on 50 Hz was applied, and then, the signals were rectified and normalized to the maximum value to Root Means Square (RMS)-EMG amplitude recorded during MVC. Moreover, the RMS during side-cutting maneuvers was measured at 50, 100, 150, and 200 milliseconds immediately after initial foot contact (i.e., when time is 0 s) [23]. We then evaluated the EMG activity of two muscles (rectus femoris and biceps femoris) during side-cuttings, before and after the 12-week period. Regarding SC30, in anterior musculature, reliability was found with reliability measures (ICC) levels of 0.78, 0.71, 0.79, and 0.82 for RMS50, RMS100, RMS150, and RMS200, respectively. In the posterior musculature, ICC of 0.72, 0.78, 0.70, and 0.77 for RMS50, RMS100, RMS150, and RMS200 was found, respectively. Regarding SC45, in anterior musculature, reliability was found with ICC levels of 0.80, 0.83, 0.71, and 0.72 for RMS50, RMS100, RMS150, and RMS200, respectively. In the posterior musculature, ICC of 0.79, 0.80, 0.83, and 0.69 for RMS50, RMS100, RMS150, and RMS200 was found, respectively. Regarding SC45cl, in anterior musculature, reliability was found with ICC levels of 0.72, 0.71, 0.71, and 0.75 for RMS50, RMS100, RMS150, and RMS200, respectively. In the posterior musculature, ICC of 0.82, 0.81, 0.72, and 0.75 for RMS50, RMS100, RMS150, and RMS200 was found, respectively.

### 2.6. Statistical Analysis

To ensure proper researcher blinding and subject anonymity, data was extracted with no names and categorical data were performed by IBM SPSS Statistics (SPSS 22.0, Chicago, IL, USA). To determinate parametricity data, a Kolmogorov–Smirnov test was examined. Moreover, a *t*-test for independent measures and repeated measures two-way ANOVA (time x group) were used to baseline data and calculate statistical differences, respectively. The significance level was adjusted with post hoc, applying Bonferroni correction, in order to minimize the likelihood of a type I error or mass significance phenomenon. Values were shown as mean ± SD and ICC were calculated. Moreover, the effect size (ES, 90% confidence limit [90%CL]) in the variables was calculated. A qualitative evaluation of possible higher or lower quantitative changes after the program was carried out as follows: <1%, almost certainly not; 1–5%, very unlikely; 5–25%, unlikely; 25–75%, possible; 75–95%, likely; 95–99%, very likely; >99%, almost certain [24].

## 3. Results

The descriptive characteristics are depicted in Table 1. No significant differences in physical activity were observed between groups (*p* > 0.05).

### 3.1. EMG in SC30

The intragroup changes produced in the EMG parameters in the EG and CG were evaluated during the SC30 (Table 2). The EG experienced a decrease in the activation of the anterior musculature (d = 9.2, *p* = 0.016). Regarding posterior musculature activation, RMS50 (1.7% [90%CL: −4.4; 8.1]), RMS100 (3.3% [90%CL: −3.2; 10.2]), RMS150 (1.8% [90%CL: −4.8; 8.8]) and RMS200 (1.6% [90%CL: −4–7; 8.3]) were reduced. In addition, all co-activation ratios were higher after intervention due to the program (H/QMVC ratio, 12.24% [90%CL: 3.9; 21.6], H/Q50 ratio, 16. 8% [90%CL: 12.1; 21.6], H/Q100 ratio, 22.5% [90%CL: 13.5; 32.1], H/Q150 ratio, 16.9% [90%CL: 6.9; 27.9] and H/Q200 ratio 20.1% [90%CL: 10.8; 30.1]).

In addition to the intragroup changes, the intergroup differences after the training program during SC30 were studied (Table 2). All activations in the anterior thigh musculature were reduced due to the training program in the EG with respect to the CG (RMSMVC, −9.2% [90%CL: −20.1; 3.2], RMS50, −10.9% [90%CL: −21.4; 1.0], RMS100, −7.7% [90%CL: −15.6; 1.0], RMS150, −7.2% [90%CL: −17.9; 4.8], and RMS200 −6.2% [90%CL: −15.3; 3.8]). The activation in the posterior thigh musculature was also less due to the training program (RMSMVC (−5.2% [90%CL: −13.8; 4.2], RMS50 (−9.5% [90%CL: −16.7; −1.6]), RMS150 (−10.9% [90%CL: −18.7; −2.4]). Finally, the effect of the training program on the H/Q50 (5.3% [90%CL: −10.8; 24.3], *p* = 0.043) and H/Q200 (−1.2% [−17.7; 18.6], *p* = 0.035) ratios showed significant differences between groups (*p* ≤ 0.05).

### 3.2. EMG in SC45

EMG parameters were analyzed during SC45 before and after the training program (Table 3). In all cases, activation of the anterior thigh musculature was reduced after the intervention program in EG. Regarding posterior musculature activation, RMSMVC increased likely due to the training program (5.2% [90%CL: −1.0; 11.8]). Regarding the co-activation ratio, all ratios were higher after the intervention period and the effect was due to the program (H/QMVC ratio: 11. 2% [90%CL: 3.5; 19.5]; ratio H/Q50: 16.9% [90%CL: 9.8; 24.5]; ratio H/Q100: 9.0 [90%CL: 2.8; 15.7]; ratio H/Q150: 10.6% [90%CL: 4.6; 16.9]; ratio H/Q200, 14.1% [90%CL: 5.3; 23.5]).

Comparison of EMG parameters between EG and CG during SC45 (Table 3) demonstrated that with respect to the activation of the anterior thigh musculature that the training program possibly caused a reduction in RMS50 (−0. 9% [90%CL: −7.0; 5.5]) and probably RMSMVC (−6.8% [90%CL: −17.9; 5.8]), RMS150 (−7.4% [90%CL: −16.2; 2.3]), and RMS200 (−6.3% [90%CL: −12.5; 0.4]) greater in EG compared to CG. In relation to the activation of the posterior musculature, training probably caused the increase in RMSMVC (−10.4% [90%CL: −21.1; 1.6]). In the rest of activations, RMS was reduced after the intervention in the EG compared to the CG. Finally, due to the training program, it was likely that all co-activation ratios were higher after 12 weeks in the EG, finding significant differences between both groups (75–95%, likely).

### 3.3. EMG in SC45cl

When the EMG parameters in the EG and CG during SC45cl (Table 4) were analyzed, it was observed that all activations of the quadriceps musculature decreased due to the training program (RMSMVC, −128% [90%CL: −16. 8; −8.6]; RMS50, −14.0% [90%CL: −17.6; −10.1]; RMS100, −13.9% [90%CL: −17.9; −9.6]; RMS150, −15.3% [90%CL: −19.5; −10.9]; RMS200, −14.0% [90%CL: −17.6; −10.3]). However, with respect to the posterior thigh musculature, all activations increased after training, although the effect of the program on RMSMVC activation was not clear (4.9% [90%CL: −2.7; 13.2]). Similarly, the co-activation ratio increased after the training program due to the program itself, being all very close to 1.00 (H/QMVC ratio, 27. 2% [90%CL: 22.4; 32.3]; H/Q50 ratio, 26.7% [90%CL: 20.4; 33.4]; H/Q100 ratio, 11.7 [90%CL: 10.7; 12.7]; H/Q150 ratio, 27.0% [90%CL: 20.7; 33.7]; H/Q200 ratio, 24.2% [90%CL: 20.1; 28.4]).

The differences between groups after the 12 weeks intervention during SC45cl are reported in Table 4. Although the activation of the quadriceps musculature decreased in all evaluations, the effect of the training program on RMSMVC, RMS100, and RMS200 was not clear. On the other hand, activation of the posterior thigh musculature increased due to the training program in both RMSH_MVC (−12.8% [90%CL: −23.1; −1.1]) and RMSH_200 (−12.4% [90%CL: −23.2; −0.1]).

Regarding the co-activation ratio, it was higher due to the training program in H/QMVC, H/Q50, H/Q100, and H/Q200. Furthermore, in the H/Q150 ratio the effect was likely due to training (9.4% [90%CL: 0.3; 19.3]).

## 4. Discussion

The aim of the present study was to evaluate the effect of a tailored and integrative neuromuscular training exercise program on EMG activity during side-cutting activities in young recreational athletes. One of the determining factors in the development of lower limbs injury and specifically knee injury has been explained in the literature by EMG factors [1,25]. In professional athletes, it seems clear that activation of the anterior and posterior thigh musculature is important for determination of the risk of knee injury, with the H/Q ratio with respect to MVC having been used in numerous studies [26,27,28]. However, this study applies a specific training program to a non-professional and physically active population at high risk of knee injury, to analyze the evolution in the related parameters, and a novel analysis of the H/QEXPLOSIVE ratio is provided.

It is commonly accepted that an H/Q ratio close to 0.8 could be considered as a turning point in the development of lower limb injury [27]. Although the area described below this ratio increases the likelihood of injury, the methodology of ratio calculation has been much debated in recent years. To date, the H/Q ratio has been determined with the standardized reference of activation in MVC. According to Zebis et al. [29], the H/Q ratio in relation to the MVC is not a metric that adequately captures the reality of persons engaging in dynamic sport behaviors. The argument is that during changes of direction the MVC will never be reached. Changes in direction are explosive actions, and according to Krosshaug et al. [30], knee injuries are accentuated in the first fifty milliseconds after contact with the ground, when activation levels have not yet allowed for close to maximum percentages. For this reason, it is important to research the H/QEXPLOSVEratio, which focuses on the initial stages of the action. Consequently, the MVC within the first 50 milliseconds (ms) of the maximum explosive contraction action was used to study the H/Q ratio in a traditional and explosive manner.

Regarding anterior musculature activation, during SC30, the EG experienced a significant decrease due to the training program (RMSMVC, d = −0.73; RMS50, d = −0.86; RMS100, d = −0.82; RMS150, d = −0.54; RMS200, d = −0.62). However, the effect on the activation of the posterior thigh musculature was not clear in any of the cases. This reduction in the activation of the quadriceps musculature in the EG could be due to enhanced timing of co-activation of the anterior and posterior thigh musculature [31]; this decrease in quadriceps activation could also explain a lower tibial translation in the critical moments of the sporting gesture and, hence, a reduction in the tension supported by the ACL [32]. A similar circumstance was found during SC45 in which EG decreased the activation of the anterior musculature while maintaining the posterior musculature at least during the first 200 ms, both being very even. However, CG decreased both quadriceps and hamstring activation. One possible explanation may be that the training program had focused the last weeks on improvement of neuromuscular control over the change of direction by means of unstable surfaces, coordination work, and combination of stimuli, which indicates a better neuromuscular control of the CG [33].

Despite control of the independent activation of the anterior and posterior thigh musculature, it seems evident that their interrelation evaluated through the H/Q ratio will provide more useful information in joint injury processes [34]. Analysis of the H/QMVC ratio during SC30 showed there were no significant differences before and after the training program in both groups, therefore the impact of the training program was unclear (d = 0.06). Although similar results were found by Ruas et al. [35], the authors differentiated several types of training and their incidence on factors associated with knee injury, which indicated that neuromuscular training might have an impact on the H/Q ratio. In the present case, a lack of specificity of the analysis strategy employed may be the cause since, as previously indicated, the traditional ratio might not be too sensitive in explosive actions. This fact is reinforced by observation of the improvements obtained in H/Q50 (0.43 [−0.95; 1.81]) and H/Q100 (1.00 [−0.09; 2.08]) in the EG after the training program. In addition, in the rest of the ratios, the CG manifested values close to 0.8, so the risk of injury in this group increased. A possible response to the current results is that reported by Greco et al. [36], who give the explosive ratio a greater involvement and suitability with respect to the traditional ratio, finding significant differences between the two. Regarding SC45 and SC45cl, all ratios improved significantly in the EG and were very close to 0.8 in the GC after 12 weeks, probably due to the training program. According to Zebis et al. [29], an improvement in these ratios could potentially indicate protection against knee injuries during changes of direction. In short, it seems that in the first instants (50 and 100 ms) of the action, which is the most compromised for the joint [23], co-activation strategies are necessary and should be trained; for this reason, the EG trained the technical movements that would later be evaluated through a specific protocol to improve proprioception and neuromuscular coordination which, as observed, resulted in an improvement in co-activation during the technical gestures analyzed. In fact, the co-activation can be explained due to an improvement in the neuromuscular mechanism during the dynamic knee action, assuming an adaptation in the neuromuscular control strategy by the EG subjects [37]. According to Zebis et al. [28], posterior musculature activation describes an elemental role in reduction of the risk of injury, as it is this musculature that usually takes longer to activate than the quadriceps musculature. In the first 50, 100, 150, and 200 ms, the ratio was significantly higher (closer to 1) after 12 weeks of training in the EG.

Concentric and eccentric work played an essential role in the current program, as they did in previous studies [38]. These authors reported that after seven weeks, the knee moment was reduced, and the co-activation of the anterior and posterior thigh musculature was improved. In the present case, after the training program, co-activation improved. In line with these findings, Asklig et al. [39] examined the eccentric component of strength and observed increased activation of the hamstring musculature, resulting in a reduction in lower limb injuries. The current study’s findings further support this phenomenon by demonstrating superior neuromuscular control, characterized by a co-activation close to 1 in the experimental group (EG), as well as increased hamstring activation during directional changes.

The training regimen utilized included a proprioceptive component to influence the neuromuscular improvement. The 12 weeks intervention included this outcome, much like Vanmeerhaeghe et al. [40] The authors of the study also discussed progression criteria that employed repetitions, visual system boundaries, and difficulty sequencing. In their research, they reported enhancements in neuromuscular control that resulted in a significant reduction in knee injuries.

In an innovative approach, the current program also combined WBV training, in line with previous studies [41]. In these studies, an improvement in knee stabilization that can be attributed to a neuromuscular improvement was also observed. One of the differences that lies between both interventions is that in the present study the vibratory work lasted for the first seven weeks and, in the previous cases, the intervention was acute (six attempts in a single day).

Finally, the balance and coordination components were also addressed in the training program to influence another of the key components of neuromuscular control. According to Begalle et al. [42], balance and coordination are crucial for enhancement of neuromuscular function. Mandelbaum et al. [43] added balance and coordination exercises in their preventive program in 1041 subjects belonging to 52 soccer teams. As a main result, they found an 88% reduction in knee problems, compared to 1905 who did not train. This training program had a positive impact on the sensorimotor system, which resulted in reduced knee injuries via enhanced neuromuscular control. The current results support this phenomenon, since both balance and coordination content were present in the training program and, as could be seen, the EG improved co-activation in both changes of direction.

## 5. Limitations and Future Research

Although the present study is novel and aimed to evaluate the effect of a tailored and integrative neuromuscular training exercise program on EMG activity during side-cutting activities in young recreational athletes, it is not without limitations. Regarding the sample, our study evaluated a sample of 57 subjects which, although the sample size was calculated, future interventions could evaluate the effect of the training program in a larger sample. Moreover, in terms of gender distribution, the sample is not homogenous.

On the other hand, in the present investigation, we evaluated numerous EMG variables during three changes of direction. The large number of variables could lead to the accumulation of Alpha errors. One response to this phenomenon has been to present a Bonferroni correction test. Future studies in this line could contemplate the evaluation of a more exhaustive description of the variables, thus mitigating the possible effect of the accumulation of a high number of variables that explain the neuromuscular phenomenon during the changes of direction.

The use of planned cutting tasks instead of unanticipated movements in the study could be perceived as a limitation. While planned cutting tasks provide valuable insights into specific aspects of neuromuscular control, they may not fully capture the sports-specific conditions and unanticipated actions associated with the risk of knee injury.

Finally, future research should be directed towards a randomized controlled trial (RCT) intervention on athletes to explain the reduction in knee injury risk due to the training program.

## 6. Conclusions

To our knowledge, this is the first study to assess the effect of a training program on EMG variables during three changes of direction in non-professional athletes. Evaluating the H/Q ratio and controlling the RMS of the anterior and posterior thigh musculature may help professionals to explain the neuro-muscular response to changes of direction. Therefore, a training program is proposed that could serve as an optimal strategy for reducing the risk of knee injuries.

## Figures and Tables

**Figure 1 sports-11-00190-f001:**
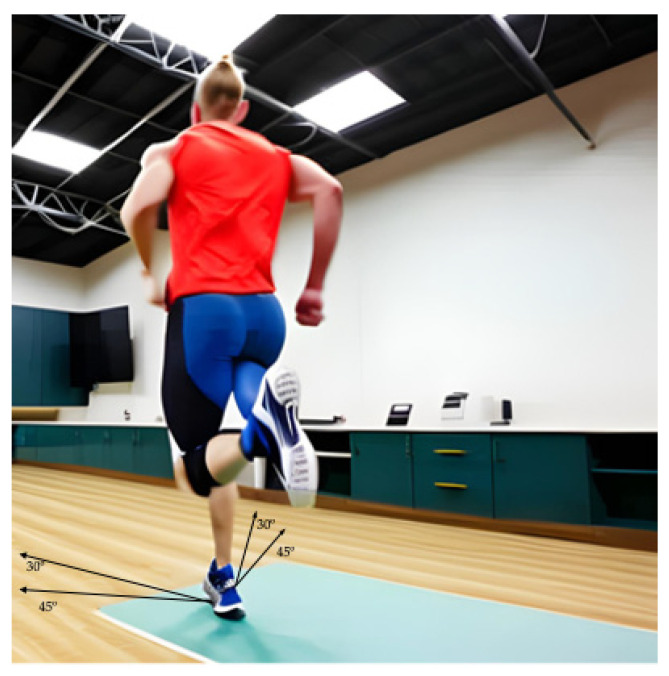
Simulation of the side-cutting tasks performed during the study.

**Table 1 sports-11-00190-t001:** Characteristics of the groups.

Variables	Mean	Men (n = 44)	Women (n = 13)	*p*	EG (n = 29)	CG (n = 28)	*p*
Age (years)	23.01 (±2.99)	23.43 (±3)	21.61 (±2)	0.358	22.3 (2.62)	23.8 (3.20)	0.056
Body mass (Kg)	73.01 (±12.76)	77.74 (±9.89)	57.01 (±7.05)	0.357	72.2 (11.1)	73.8 (14.5)	0.638
Height (m)	1.74 (±0.08)	±1.78 (±0.05)	1.63 (±0.06)	0.600	1.74 (0.08)	1.76 (0.08)	0.266
BMI (kg/m^2^)	23.74 (±2.71)	24.46 (±2.49)	21.28 (±1.90)	0.383	23.9 (2.51)	23.6 (2.95)	0.708
PA (hours/week)	8.78 (±4.30)	9.30 (±4.49)	7.00 (±3.13)	0.251	8.33 (4.28)	9.25 (4.37)	0.424
PA_day_ (hours/day)	2.32 (±2.76)	2.25 (±2.43)	2.57 (±4.78)	0.303	2.40 (2.95)	2.25 (2.61)	0.843

Data are reported as Mean (±SD); BMI: body mass index; PA: physical activity; Kg = kilograms; m = meters. *p* ≤ 0.05.

**Table 2 sports-11-00190-t002:** Intra- and inter-group changes on EMG during SC30, mean ±SD.

Variables	Pre		Post		Differences		
	EG	CG	EG	CG	*p* Value ^a^	d-Cohen	Change
RMS_Q_MVC_	0.6426 (0.04)	0.6511 (0.08)	0.5723 (0.07)	0.6351 (0.09)	0.016	−9.2 (−20.1; 3.2)	5/13/82%
RMS_H_MVC_	0.5613 (0.04)	0.5740 (0.05)	0.5571 (0.15)	0.5823 (0.08)	0.197	−5.2 (−13.8; 4.2)	5/23/72%
RMS_Q_50_	0.6526 (0.03)	0.6565 (0.17)	0.5541 (0.13)	0.6486 (0.08)	0.121	−10.9 (−21.4; 1.0)	3/9/88%
RMS_H_50_	0.5435 (0.03)	0.5059 (0.11)	0.5282 (0.17)	0.5632 (0.03)	0.785	−9.5 (−16.7; −1.6)	1/3/95%
RMS_Q_100_	0.6534 (0.03)	0.6574 (0.16)	0.5670 (0.13)	0.6417 (0.09)	0.251	−7.7 (−15.6; 1.0)	4/10/87%
RMS_H_100_	0.5502 (0.04)	0.5320 (0.11)	0.5432 (0.18)	0.5582 (0.04)	0.973	−0.2 (−9.7; 10.2)	38/20/41%
RMS_Q_150_	0.6434 (0.05)	0.6723 (0.17)	0.5365 (0.13)	0.6440 (0.08)	0.038	−7.2 (−17.9; 4.8)	4/19/75%
RMS_H_150_	0.5478 (0.03)	0.5208 (0.11)	0.5337 (0.18)	0.5618 (0.04)	0.463	−10.9 (−18.7; −2.4)	1/3/96%
RMS_Q_200_	0.6455 (0.04)	0.6907 (0.16)	0.5398 (0.13)	0.6487 (0.08)	0.012	−6.2 (−15.3; 3.8)	5/19/76%
RMS_H_200_	0.5431 (0.03)	0.5281 (0.12)	0.5475 (0.15)	0.5538 (0.03)	0.949	4.9 (−4.3; 14.9)	70/19/11%
Ratio_H/Q_MVC_	0.8770 (0.09)	0.8642 (0.06)	0.9851 (0.28)	0.9330 (0.17)	0.861	−0.5 (−12.6; 13.3)	38/19/43%
Ratio_H/Q_50_	0.8253 (0.05)	0.7701 (0.09)	1.0031 (0.28)	0.8440 (0.39)	0.043	5.3 (−10.8; 24.3)	61/17/22%
Ratio_H/Q_100_	0.8446 (0.08)	0.8232 (0.12)	0.9714 (0.31)	0.7800 (0.35)	0.013	14.1 (−1.2; 31.7)	89/8/4%
Ratio_H/Q_150_	0.8583 (0.10)	0.7732 (0.08)	0.9714 (0.31)	0.8332 (0.38)	0.024	0.0 (−18.3; 22.4)	41/19/41%
Ratio_H/Q_200_	0.8459 (0.09)	0.7762 (0.12)	0.9852 (0.29)	0.8398 (0.39)	0.035	−1.2 (−17.7; 18.6)	35/22/43%

*p* < 0.05. ^a^ Adjustment for multiple comparisons: Bonferroni correction. RMS (mv) = root means square; Q = quadriceps; H = hamstrings; MVC = Maximum Voluntary Contraction; 50/100/150/200 (ms) = in the first 50/100/150/200 milliseconds; Ratio_H/Q_ = ratio of co-activation.

**Table 3 sports-11-00190-t003:** Intra- and inter-group changes on EMG during SC_45_, mean ±SD.

Variables	Pre		Post		Differences		
	EG	CG	EG	CG	*p* Value ^a^	d-Cohen	Change
RMS_Q_MVC_	0.6543 (0.02)	0.6708 (0.06)	0.5453 (0.07)	0.6161 (0.14)	0.044	−6.8 (−17.9; 5.8)	5/7/88%
RMS_H_MVC_	0.5554 (0.05)	0.5422 (0.05)	0.5974 (0.11)	0.6287 (0.12)	0.418	−10.4 (−21.1; 1.6)	5/6/89%
RMS_Q_50_	0.6553 (0.02)	0.6501 (0.02)	0.5285 (0.15)	0.5151 (0.24)	0.585	−0.9 (−7.0; 5.5)	5/28/67%
RMS_H_50_	0.5602 (0.05)	0.5414 (0.03)	0.5477 (0.18)	0.5222 (0.23)	0.497	−10.8 (−20.7; 0.4)	4/4/92%
RMS_Q_100_	0.6471 (0.02)	0.6524 (0.02)	0.5140 (0.15)	0.5051 (0.24)	0.843	5.3 (−3.0; 14.4)	81/8/11%
RMS_H_100_	0.5575 (0.05)	0.5413 (0.04)	0.5396 (0.18)	0.5107 (0.23)	0.488	−11.0 (−24.1; 4.3)	5/10/85%
RMS_Q_150_	0.6544 (0.02)	0.6611 (0.02)	0.4954 (0.15)	0.5036 (0.24)	0.781	−7.4 (−16.2; 2.3)	5/8/88%
RMS_H_150_	0.5546 (0.05)	0.5491 (0.04)	0.5254 (0.17)	0.5212 (0.23)	0.917	−7.6 (−20.8; 7.8)	4/20/76%
RMS_Q_200_	0.6554 (0.02)	0.6589 (0.02)	0.4968 (0.14)	0.5137 (0.24)	0.795	−6.3 (−12.5; 0.4)	4/5/91%
RMS_H_200_	0.5610 (0.05)	0.5395 (0.04)	0.5154 (0.17)	0.5091 (0.23)	0.712	−15.2 (−26.0; −3.0)	2/2/96%
Ratio_H/Q_MVC_	0.8398 (0.05)	0.8143 (0.10)	0.9504 (0.18)	0.8512 (0.10)	0.836	15.9 (2.0; 31.6)	95/3/2%
Ratio_H/Q_50_	0.8432 (0.05)	0.8339 (0.06)	0.9958 (0.15)	0.8440 (0.10)	0.186	13.3 (−0.5; 29.0)	92/4/4%
Ratio_H/Q_100_	0.8493 (0.05)	0.8303 (0.06)	0.9382 (0.14)	0.8589 (0.08)	0.263	7.2 (−3.7; 19.3)	80/14/5%
Ratio_H/Q_150_	0.8368 (0.05)	0.8318 (0.07)	0.9368 (0.15)	0.8501 (0.11)	0.327	8.2 (−4.7; 22.8)	79/15/4%
Ratio_H/Q_200_	0.8459 (0.05)	0.8200 (0.07)	0.9774 (0.17)	0.8586 (0.10)	0.298	11.5 (−2.1; 26.9)	88/7/5%

*p* < 0.05. ^a^ Adjustment for multiple comparisons: Bonferroni correction. RMS (mv) = root Mmeans square; Q = quadriceps; H = hamstrings; MVC = Maximum Voluntary Contraction; 50/100/150/200 (ms) = in the first 50/100/150/200 milliseconds; Ratio_H/Q_ = ratio of co-activation.

**Table 4 sports-11-00190-t004:** Intra- and inter-group changes on EMG during SC_45cl_, mean ±SD.

Variables	Pre		Post		Differences		
	EG	CG	EG	CG	*p* Value ^a^	d-Cohen	Change
RMS_Q_MVC_	0.6721 (0.03)	0.6123 (0.04)	0.5762 (0.08)	0.6209 (0.14)	0.409	−3.3 (−13.4; 8.0)	25/11/64%
RMS_H_MVC_	0.5267 (0.05)	0.5976 (0.04)	0.5865 (0.13)	0.6357 (0.12)	0.203	−12.8 (−23.1; −1.1)	2/4/94%
RMS_Q_50_	0.6724 (0.03)	0.5942 (0.05)	0.5748 (0.07)	0.6008 (0.18)	0.312	−4.0 (−14.7; 8.0)	22/11/66%
RMS_H_50_	0.5225 (0.05)	0.5833 (0.03)	0.5855 (0.12)	0.6143 (0.17)	0.713	−10.8 (−23.1; 3.4)	6/8/85%
RMS_Q_100_	0.6698 (0.03)	0.6012 (0.04)	0.5707 (0.08)	0.5855 (0.18)	0.568	−2.9 (−14.5; 10.1)	29/11/60%
RMS_H_100_	0.5206 (0.05)	0.5419 (0.03)	0.6054 (0.11)	0.6144 (0.17)	0.738	−10.6 (−22.0; 2.4)	5/8/87%
RMS_Q_150_	0.6713 (0.03)	0.6574 (0.05)	0.5644 (0.08)	0.5819 (0.18)	0.503	−5.2 (−16.9; 8.1)	20/10/70%
RMS_H_150_	0.5297 (0.04)	0.5743 (0.04)	0.5901 (0.12)	0.6078 (0.17)	0.682	−11.3 (−23.1; 2.4)	6/6/88%
RMS_Q_200_	0.6686 (0.03)	0.6686 (0.02)	0.5660 (0.07)	0.5939 (0.18)	0.328	−2.6 (−14.0; 10.3)	30/11/58%
RMS_H_200_	0.5229 (0.05)	0.5739 (0.05)	0.5989 (0.11)	0.6164 (0.17)	0.947	−12.4 (−23.2; −0.1)	3/5/92%
Ratio_H/Q_MVC_	0.7847 (0.08)	0.8007 (0.05)	10.04 (0.08)	0.7869 (0.03)	0.171	25.8 (17.4; 34.8)	100/0/0%
Ratio_H/Q_50_	0.7862 (0.08)	0.8125 (0.04)	10.02 (0.11)	0.7979 (0.03)	0.218	20.0 (10.0; 31.0)	100/0/0%
Ratio_H/Q_100_	0.8712 (0.01)	0.8544 (0.03)	0.9607 (0.01)	0.8381 (0.04)	0.201	15.9 (12.2; 19.6)	100/0/0%
Ratio_H/Q_150_	0.7911 (0.07)	0.8234 (0.06)	10.02 (0.14)	0.8499 (0.04)	0.269	9.4 (0.3; 19.3)	91/7/2%
Ratio_H/Q_200_	0.7839 (0.09)	0.7877 (0.06)	10.01 (0.08)	0.8065 (0.03)	0.351	20.4 (11.2; 30.3)	100/0/0%

*p* < 0.05. ^a^ Adjustment for multiple comparisons: Bonferroni correction. RMS (mv) = root means square; Q = quadriceps; H = hamstrings; MVC = Maximum Voluntary Contraction; 50/100/150/200 (ms) = in the first 50/100/150/200 milliseconds; Ratio_H/Q_ = ratio of co-activation.

## Data Availability

Not Applicable.

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
