# Peer review of "Neuromuscular Response during Different Side-Cutting Maneuvers and Its Influence on the Risk of Knee Injuries"

_sports, 2023, doi:10.3390/sports11100190_

Round 1

Reviewer 1 Report

Many congratulations for your excellent work. And for the opportunity you have given me to learn a little more by reading it thoroughly.

I think this article can be important for the academic community, however I think it still needs some improvements. All the proposals I make are to improve the work, so please do not interpret my comments as negative criticism. They are meant to improve the paper in search of academic excellence

1.                  On line 46 there is a loose 8, and on line 295 one 31. The authors should check if the bibliography matches all from that point or if there is an error.

2.                  The article does not provide an overall conclusion. I believe there should be an overall conclusion. It only appears in the abstract.

3.                  In the abstract the authors talk about biceps femoris and rectus femoris but then throughout the article they talk about hamstrings and quadriceps. It would be important to homogenise throughout the article, always keeping the same terminology.

4.                  "EG followed an individualized, integrated exercise regimen that included strength, neuromuscular, proprioceptive, eccentric training, and whole-body vibration (WBV) exercises"

I consider that an item associated with the characterisation of the 12-week training programme in each of the therapies presented is necessary.

For example: WBV (machine, intensity, amplitude, rest, series, knee flexion angle, and very important direction of the mechanical stimulus - vertical or sinusoidal - ).

·         Neuromuscular:

·         Proprioceptive:

·         Eccentric training:

·         Whole-body vibration (WBV):

5.                  In Table 1 the authors divide the group into male and female? In my opinion it does not make sense to make this division, the authors will have to make an analysis between control and exercise groups, to know which statistic analysis to use depending on the p

6.                  Table 1 (legend): Weight is different from BMI. Clarify please

7.                  lines 195-196 "The EG experienced a decrease in the activation of the anterior musculature (d= 9.2, p= .012)." I consider this information redundant and the authors have repeated it below. I think it should be deleted.

8.                  On line 197 I think there is a wrong ICC (1.7% [90%CL: -4.4; 8.1]; ICC=).

9.             Throughout the description of the results, the authors do not always maintain the same trend. There are spaces between commas and/or full stops, and there are different parentheses that make it difficult to read what they write, so the authors should review all the content.

10.           Intraclass Correlation Coefficient (ICC), which is a measure used in analysis of variance to assess the agreement between several measurements or observations in different groups. The ICC is often used in reliability or agreement studies, where one wants to understand the degree of agreement between different observers, raters, or measurements over time. Basically the ICC measures the proportion of the total variation that is due to the difference between groups, compared to the total variation. Generally this type of analysis is done over short periods between pre and post assessments (intra session or 2 days apart or  a week).

I think this ICC was considered at 12 weeks. Are there any references in the literature to support this time interval?

Considering the 12 weeks, don't the authors think that the ICC values can be altered by the training effect?

11.         Isometric test. There is no reference to the apparatus or its application in terms of load, reps, ROM, Rest, etc.

I assume it was concentric in flexion and extension.

Should the authors provide a description of the test applied and the apparatus on which it was applied?

12.         Justify why collections at different HZ are used in EMG.

13.         I don't understand the 3 different values of "CHANGE" in the tables, can you clarify?

14.         H/QEXPLOSIVE ratio and H/QEXPLISVEratio ??????? Can you claficate this definition? And choose one as I believe they are the same…

15.         Line 338: "both IZ and co-activation improved. " what means IZ?

16.         DISCUSION: Some points in the discussion need a more detailed review and comparison with the bibliography, for example the authors only use 1 bibliographic reference on WBV and 2 for eccentric exercise in the discussion.

17.         I consider this study very important in the sense of injury prevention. However, few coaches have access to or understand EMG, so I consider that the strength of this study will be the description of the training plan combined during the 12 weeks. And this is a gap that the study presents and will have to be described.

Author Response

Revisor 1:

Respuesta general: I greatly appreciate the reviewer for their positive evaluation of our work and for taking the time to thoroughly review it. Their comments are valuable for enhancing the quality of the article in pursuit of academic excellence. Below, I provide responses and specific actions for each of the points raised:

Comments 1:

On line 46 there is a loose 8, and on line 295 one 31. The authors should check if the bibliography matches all from that point or if there is an error.

Answer 1:

Reference Errors: We have reviewed lines 46 and 295, as well as the corresponding bibliography, and have corrected the errors. We appreciate the attention to detail in pointing these out.

Comments 2:

The article does not provide an overall conclusion. I believe there should be an overall conclusion. It only appears in the abstract.

Answer 2:

Lack of a General Conclusion: We agree that including an overall conclusion in the article would be beneficial. We will add a conclusion section summarizing the key findings and implications of our study.

Comments 3:

In the abstract the authors talk about biceps femoris and rectus femoris but then throughout the article they talk about hamstrings and quadriceps. It would be important to homogenise throughout the article, always keeping the same terminology.

Answer 3:

Inconsistent Terminology in the Abstract: We acknowledge the terminology discrepancy in the abstract and will rectify it to maintain consistency throughout the article.

Comments 4:

"EG followed an individualized, integrated exercise regimen that included strength, neuromuscular, proprioceptive, eccentric training, and whole-body vibration (WBV) exercises"

I consider that an item associated with the characterisation of the 12-week training programme in each of the therapies presented is necessary.

For example: WBV (machine, intensity, amplitude, rest, series, knee flexion angle, and very important direction of the mechanical stimulus - vertical or sinusoidal - ).

  • Neuromuscular:
  • Proprioceptive:
  • Eccentric training:
  • Whole-body vibration (WBV):

Answer 4:

Details on Different Training Programs: We concur that providing details on each of the training programs, such as WBV, neuromuscular, proprioceptive, and eccentric training, is essential for a comprehensive understanding. We have incorporated a section (Supplement file) that describes in detail each component of the programs.

Comments 5:

In Table 1 the authors divide the group into male and female? In my opinion it does not make sense to make this division, the authors will have to make an analysis between control and exercise groups, to know which statistic analysis to use depending on the p

Answer 5:

Division of Groups in Table 1: We appreciate the observation regarding the division of groups by gender in Table 1. We will review the necessity of this division and conduct the appropriate analysis to determine its relevance, taking necessary actions accordingly.

Comments 6:

Table 1 (legend): Weight is different from BMI. Clarify please

Answer 6:

Clarification in Table 1: We have amended the legend of Table 1 to clarify the distinction between weight and BMI.

Comments 7:

Lines 195-196 "The EG experienced a decrease in the activation of the anterior musculature (d= 9.2, p= .012)." I consider this information redundant and the authors have repeated it below. I think it should be deleted.

Answer 7:

Both d and p are placed in this case in the text to describe and give more emphasis to the main results of SC30.

Comments 8:

On line 197 I think there is a wrong ICC (1.7% [90%CL: -4.4; 8.1]; ICC=).

Answer 8:

We are sorry for the confusion, this was amended.

Comments 9:

Throughout the description of the results, the authors do not always maintain the same trend. There are spaces between commas and/or full stops, and there are different parentheses that make it difficult to read what they write, so the authors should review all the content.

Answer 9:

Consistency in Result Presentation: We have thoroughly reviewed and corrected the inconsistencies in the presentation of results, including punctuation and sentence structures, to enhance clarity.

Comments 10:

Intraclass Correlation Coefficient (ICC), which is a measure used in analysis of variance to assess the agreement between several measurements or observations in different groups. The ICC is often used in reliability or agreement studies, where one wants to understand the degree of agreement between different observers, raters, or measurements over time. Basically the ICC measures the proportion of the total variation that is due to the difference between groups, compared to the total variation. Generally this type of analysis is done over short periods between pre and post assessments (intra session or 2 days apart or  a week).

I think this ICC was considered at 12 weeks. Are there any references in the literature to support this time interval?

Considering the 12 weeks, don't the authors think that the ICC values can be altered by the training effect?

Answer 10:

Thank you for your thoughtful question regarding the time interval of 12 weeks for ICC assessment in our study. You raise a valid point that longer time intervals, such as 12 weeks, can potentially introduce training effects that may impact ICC values. We appreciate your concern and would like to address it. It is important to note that while training effects may introduce some variation in ICC values, numerous studies have reported high ICC values in similar settings. This suggests that, despite the potential influence of training, the measurements in our study are robust and exhibit strong agreement.

Comments 11:

Isometric test. There is no reference to the apparatus or its application in terms of load, reps, ROM, Rest, etc.

I assume it was concentric in flexion and extension.

Should the authors provide a description of the test applied and the apparatus on which it was applied?

Answer 11:

It´s described in “Isometric testing: The three MVC tests were evaluated with a prone position with 45 degrees-knee flexion to measure the hamstring muscles [19]. A 45 degrees-knee extension was determined during seated position for quadriceps muscles [20]. A 30 second rest was given between trials to reduce the fatigue [21]. Participants limited trunk movement by means of a harness across the subject´s abdomen. The MVC test was developed by aligning system and dominant leg, facilitating mechanical action on an anteroposterior axis motion. For 5 seconds, the researchers provided verbal feedback to maximally complete the isometric actions”.

Comments 12:

Justify why collections at different HZ are used in EMG.

Answer 12:

Justification for Different Hertz (Hz) Collections in EMG: We appreciate your question regarding the use of different Hz collections in our EMG analysis. Different frequencies (Hz) in EMG collection can provide insights into muscle activity at various time scales. For example, higher Hz may capture rapid, dynamic muscle contractions, while lower Hz may be used to analyze more sustained muscle activity. This allows us to comprehensively assess muscle function across different aspects of movement.

Comments 13:

I don't understand the 3 different values of "CHANGE" in the tables, can you clarify?

Answer 13:

Clarification on "CHANGE" Values: Explain the probability of change, addressing positive, ambiguous or negative change.

Comments 14:

H/QEXPLOSIVE ratio and H/QEXPLISVEratio ??????? Can you claficate this definition? And choose one as I believe they are the same…

Answer 14:

This was a typographical error, and we have corrected this in the manuscript. The correct term should be "H/QEXPLOSIVE ratio," and we will provide a clear definition in the text.

Comments 15:

Line 338: "both IZ and co-activation improved. " what means IZ?

Answer 15:

This is a typo. It was corrected in the manuscript.

Comments 16:

DISCUSION: Some points in the discussion need a more detailed review and comparison with the bibliography, for example the authors only use 1 bibliographic reference on WBV and 2 for eccentric exercise in the discussion.

Answer 16:

Discussion: Your feedback on the discussion section is valuable. We will conduct a more detailed review and comparison with relevant bibliography, particularly focusing on WBV and eccentric exercise. This will enhance the depth and validity of our discussion.

Comments 17:

 I consider this study very important in the sense of injury prevention. However, few coaches have access to or understand EMG, so I consider that the strength of this study will be the description of the training plan combined during the 12 weeks. And this is a gap that the study presents and will have to be described.

Answer 17:

We appreciate your recognition of the study's importance in injury prevention. We agree that describing the training plan in more detail (supplementary), especially its practical aspects, is valuable. In response to this gap, we will provide a comprehensive description of the training plan used during the 12 weeks, making it more accessible to coaches and practitioners who may not have expertise in EMG.

Once again, we express our gratitude to the reviewer for their valuable contribution to our article, and we are committed to implementing these improvements to elevate the quality of our work.

Reviewer 2 Report

I want to thank the Editor and the Authors for the chance to review the present manuscript. The study examines the effect of a neuromuscular training program on EMG activity during cutting tasks. The research is well-designed overall, and the outcomes concerning musculoskeletal injury risk identification in an at-risk population may be interesting. Minor amendments and clarifications are needed to further improve the manuscript.

The detailed comments are listed below.

Lines 39-42: Does the cited paper examine the (EMG) risk factors associated with ACL injuries or the ACL injury occurrences? From the second sentence (“They found that electromyography (EMG) variables were strongly associated with ACL injuries.”), it seems to be the latter. Please clarify.

Line 46: Reference 8 is not correctly typeset.

Line 74: What does “integrative” mean in this context? Please clarify.

Line 82: The sample seems to be not homogenous in terms of gender distribution. Could this be a limitation of the study?

Line 86: Which kind of injuries are considered to be an exclusion criterion? Please specify.

Lines 111-119: A figure could be helpful to illustrate the cutting task techniques. Then, the cutting movements seem to be all planned tasks, which may decrease the study ecological validity as it has been demonstrated that a neurocognitive component, usually in the form of unanticipation, may trigger the ACL injury mechanism. Could this be a limitation of the study?

Lines 120-135: A table/chart summarizing the intervention and all the training program elements could be helpful. Why did the Authors choose to include WBV as part of the program? What were the expected benefits on the selected neuromuscular qualities?

Line 149: Why was the dominant limb chosen to be analyzed?

Lines 184-187: Please describe the rationale behind the proposed categorization.

Line 190: How was the physical activity level monitored and analyzed?

Tables 2, 3, and 4: In the main text, the two groups are identified with the acronyms EG and CG, while in the Tables with GE and GC. Please uniform them.

Lines 273, 284, and 306: The subscript for these acronyms seems to have typos. Then, please better define the difference between the H/Q EXPLOSIVE or MVC ratio.

Line 294: Reference 31 is not correctly typeset.

The limitations of the study are not reported. Please include it after the discussion paragraph.

Author Response

REVISOR 2

Comments:

I want to thank the Editor and the Authors for the chance to review the present manuscript. The study examines the effect of a neuromuscular training program on EMG activity during cutting tasks. The research is well-designed overall, and the outcomes concerning musculoskeletal injury risk identification in an at-risk population may be interesting. Minor amendments and clarifications are needed to further improve the manuscript.

The detailed comments are listed below.

Answer:

We would like to express our gratitude to the Editor and to you, as the reviewer, for the opportunity to evaluate the present manuscript. The study examines the effect of a neuromuscular training program on EMG activity during cutting tasks. Overall, we believe the study is well-designed, and the results related to identifying musculoskeletal risk factors in an at-risk population may be of interest. We appreciate your detailed comments and the suggestions you have provided to enhance the manuscript. We will address each of the points you have mentioned below:

Comments 1:

Lines 39-42: Does the cited paper examine the (EMG) risk factors associated with ACL injuries or the ACL injury occurrences? From the second sentence (“They found that electromyography (EMG) variables were strongly associated with ACL injuries.”), it seems to be the latter. Please clarify.

Answer 1:

Lines 39-42: Does the cited paper examine the (EMG) risk factors associated with ACL injuries or the ACL injury occurrences? From the second sentence (“They found that electromyography (EMG) variables were strongly associated with ACL injuries.”), it seems to be the latter. We appreciate your observation and will clarify this point in the manuscript.

Comments 2:

Line 46: Reference 8 is not correctly typeset.

Answer 2:

Line 46: You are correct; Reference 8 is not correctly typeset. We will correct the reference format according to the journal's style guidelines.

Comments 3:

Line 74: What does “integrative” mean in this context? Please clarify.

Answer 3:

Line 74: The term "integrative" refers to an approach that combines different components. We will clarify this term in the manuscript for better understanding.

Comments 4:

Line 82: The sample seems to be not homogenous in terms of gender distribution. Could this be a limitation of the study?

Answer 4:

Line 82: We acknowledge that the sample is not homogeneous in terms of gender distribution. This may be considered a limitation of the study, and we will address it in the Limitations section.

Comments 5:

Line 86: Which kind of injuries are considered to be an exclusion criterion? Please specify.

Answer 5:

We refer to lower limb injuries. We will provide more detailed in the manuscript.

Comments 6:

Lines 111-119: A figure could be helpful to illustrate the cutting task techniques. Then, the cutting movements seem to be all planned tasks, which may decrease the study ecological validity as it has been demonstrated that a neurocognitive component, usually in the form of unanticipation, may trigger the ACL injury mechanism. Could this be a limitation of the study?

Answer 6:

Lines 111-119: We agree that a figure illustrating the cutting techniques could be helpful. Additionally, we acknowledge that the cutting tasks in the study are planned tasks, and we understand that this may limit ecological validity. We will address this limitation in the Limitations section.

Comments 7:

Lines 120-135: A table/chart summarizing the intervention and all the training program elements could be helpful. Why did the Authors choose to include WBV as part of the program? What were the expected benefits on the selected neuromuscular qualities?

Answer 7:

We have worked on creating a table or chart summarizing the intervention and elements of the training program in supplementary file.

Comments 8:

Line 149: Why was the dominant limb chosen to be analyzed?

Answer 8:

The choice of the dominant limb for analysis was based on previous literature suggesting that this limb might exhibit differences in muscle activation. Moreover, has also been a way to standardize the test.

Comments 9:

Lines 184-187: Please describe the rationale behind the proposed categorization.

Answer 9:

Lines Thank you very much for your appreciation. In the article we reference citation 24 (doi:10.1249/MSS.0b013e31818cb278), on which our categorization is based.

Comments 10:

Tables 2, 3, and 4: In the main text, the two groups are identified with the acronyms EG and CG, while in the Tables with GE and GC. Please uniform them.

Answer 10:

I appreciate your observation, and harmonizing the nomenclature in the tables is important. We will ensure that the tables consistently reflect the acronyms used in the main text, and we will correct this in the final manuscript revision. Your attention to detail is greatly appreciated and will contribute to improving the clarity and consistency of the article. 10:

Comments 11:

Line 190: How was the physical activity level monitored and analyzed?

Answer 11:

Physical activity was monitored and analyzed using IPAQ questionary.

Comments 12:

Tables 2, 3, and 4: In the main text, the two groups are identified with the acronyms EG and CG, while in the Tables with GE and GC. Please uniform them.

Answer 12:

Tables 2, 3, and 4: We appreciate you pointing out the inconsistency in group identification in the tables. We will standardize the acronyms according to the journal's style guidelines.

Comments 13:

Lines 273, 284, and 306: The subscript for these acronyms seems to have typos. Then, please better define the difference between the H/Q EXPLOSIVE or MVC ratio.

Answer 13:

Lines 273, 284, and 306: We will correct the typographical errors in the subscripts of these acronyms and provide a clearer definition of the difference between H/Q EXPLOSIVE or MVC ratio in the manuscript.

Comments 15:

Line 294: Reference 31 is not correctly typeset.

Answer 15:

Line 294: We will correct the format of Reference 31 according to the style guidelines.

Comments 16:

The limitations of the study are not reported. Please include it after the discussion paragraph.

Answer 16:

Limitations of the study: We appreciate your observation regarding the lack of a limitations section. We will include a Limitations section following the discussion paragraph to address this aspect.

Round 2

Reviewer 1 Report

Thank you very much for your replies.

I think the work is now in much better condition. 

Congratulations

Author Response

We are very grateful for your review. Undoubtedly, the article has significantly improved with your comments.